# CollectiveKV: Decoupling and Sharing Collaborative Information in Sequential Recommendation

**Jingyu Li**[1,3]* **Zhaocheng Du**[2]* **Qianhui Zhu**[4]**, kaiyuan Li**[4]**, Zhicheng Zhang**[4]**, Song-Li Wu**[4]**,
Chaolang Li**[1,3]**, Pengwen Dai**[1,3]†

[1]School of Cyber Science and Technology, Shenzhen Campus of Sun Yat-sen University
[2]Huawei Noah's Ark Lab
[3]Shenzhen Key Laboratory of Adversarial Artificial Intelligence
[4]Tsinghua Shenzhen International Graduate School
`{lijy768, lichlang5}@mail2.sysu.edu.cn,`
`duzhaocheng1998@gmail.com`
`{zhuqh23, likaiyua23, zhang-zc24, wsl24}@mails.tsinghua.edu.cn`
`daipw@mail.sysu.edu.cn`

## Abstract

Sequential recommendation models are widely used in applications, yet they face stringent latency requirements. Mainstream models leverage the Transformer attention mechanism to improve performance, but its computational complexity grows with the sequence length, leading to a latency challenge for long sequences. Consequently, KV cache technology has recently been explored in sequential recommendation systems to reduce inference latency. However, KV cache introduces substantial storage overhead in sequential recommendation systems, which often have a large user base with potentially very long user history sequences. In this work, we observe that KV sequences across different users exhibit significant similarities, indicating the existence of collaborative signals in KV. Furthermore, we analyze the KV using singular value decomposition (SVD) and find that the information in KV can be divided into two parts: the majority of the information is shareable across users, while a small portion is user-specific. Motivated by this, we propose CollectiveKV, a cross-user KV sharing mechanism. It captures the information shared across users through a learnable global KV pool. During inference, each user retrieves high-dimensional shared KV from the pool and concatenates them with low-dimensional user-specific KV to obtain the final KV. Experiments on five sequential recommendation models and three datasets show that our method can compress the KV cache to only 0.8% of its original size, while maintaining or even enhancing model performance.

## 1 Introduction

Sequential recommendation models are widely used in Internet applications such as short video platforms, e-commerce, and real-time advertising, where ensuring a smooth user experience requires meeting stringent latency constraints. Existing methods usually focus on improving the accuracy of recommendations by introducing or improving the Transformer attention mechanism Vaswani et al. (2017). However, the computational cost of attention grows with the input sequence length, making it challenging to satisfy the strict latency requirements in practical deployment.

In the attention mechanism, projecting the input sequence into queries (Q), keys (K), and values (V) via linear transformations accounts for a significant portion of the computational cost. To mitigate inference latency, recent works such as HSTU Zhai et al. (2024) and MARM Lv et al. (2024) introduce the KV cache technology into recommendation systems, which precomputes and stores keys

---

*These authors contributed equally to this work.
†Corresponding author

and values for reuse during inference. However, sequential recommendation systems usually serve a massive user base, where each user may have a long behavior history Zhang et al. (2026); Ma et al. (2025); Yu et al. (2026). When user history sequences are projected into the KV cache, they will incur substantial storage overhead and introduce considerable communication latency. This motivates the need for efficient KV cache compression.

Recently, many works Shazeer (2019); Ainslie et al. (2023); Singhania et al. (2024); Tang et al. (2024); Liu et al. (2024) have attempted to reduce the KV cache size in LLMs, primarily by compressing each KV sequence through reducing its length or dimension. In contrast, sequential recommendation scenarios present a unique property: user behaviors are not independent but interrelated, exhibiting strong collaborative signals. Motivated by this, we analyze the similarities of Keys and Values across users in multiple sequential recommendation models. As shown in Figure 1, different users indeed display a certain degree of K/V similarity, suggesting that collaborative signals are also embedded in the K/V representations. This observation inspires us to explore a new compression strategy—sharing the KV cache **across users**.

To further quantify the shareable portion of K/V, we apply singular value decomposition (SVD) to decompose the keys and values into a principal component subspace and a residual subspace, denoted as principal K/V and residual K/V, respectively. The principal K/V captures the majority of the information, while the residual K/V contains only a small fraction. We then measure cross-user similarities separately on these two parts. As shown in Figure 2, principal K/V are more likely to exhibit positive or negative correlations across users, whereas residual K/V similarities cluster around zero. **This finding suggests that most information in the keys and values can be shared across users, while only a small portion remains user-specific.**

Motivated by these results, we decompose the KV into two parts: the low-dimensional user-specific KV and the high-dimensional collective KV shared across users. To be specific, we utilize a learnable global K/V pool to capture the information shared across users. In this pool, the global KVs are designed to be high-dimensional to provide greater information capacity. Each user retrieves the corresponding shared K/V—referred to as the collective K/V—from the global K/V pool, based on their own behavior sequence. On the other side, the user-specific KVs are represented as low-dimensional linear projections of each user's historical behavior sequence embeddings. Finally, the user-specific K/V and collective K/V are concatenated to formulate the final K/V. In this way, by introducing the sharing mechanism, the user-specific KV only needs to carry limited information, allowing it to be represented with just a few dimensions, thereby reducing the KV cache size. Experiments on both target attention and self-attention models demonstrate that our method achieves excellent compression while preserving or even improving model performance.

## 2 BACKGROUND AND RELATED WORK

### 2.1 KV CACHE IN SEQUENTIAL RECOMMENDATION SYSTEMS

In sequential recommendation models, target attention and self-attention are two widely used attention mechanisms. In target attention models Pi et al. (2020); Cao et al. (2022); Chen et al. (2022); Chang et al. (2023), the query $\mathbf{Q} \in \mathbb{R}^{1 \times d}$ is projected from a target item, where $d$ is the head dimension. In self-attention models Kang & McAuley (2018); Zhai et al. (2024), the query $\mathbf{Q} \in \mathbb{R}^{s \times d}$ is projected from the input sequence, where $s$ is the sequence length. In both paradigms, the key $\mathbf{K} \in \mathbb{R}^{s \times d}$ and value $\mathbf{V} \in \mathbb{R}^{s \times d}$ are projected from the input sequence, meaning that their shapes are identical and both can benefit from the KV cache mechanism.

Building on this property, recent works such as HSTU Zhai et al. (2024) and MARM Lv et al. (2024) introduce precomputing and caching the keys and values before online inference, thereby reducing online inference latency. These studies demonstrate the effectiveness of KV cache in sequential recommendation. However, unlike in LLM scenarios where the KV cache is maintained per sequence during a single inference session, recommendation systems must maintain the KV cache *per user*. Given the massive user base (often hundreds of millions) and the potentially long interaction history of each user, the aggregate KV cache size quickly exceeds GPU memory capacity. Consequently, offloading KV caches to CPU memory or external storage becomes inevitable. Yet this offloading introduces significant transfer latency between the GPU and external storage. Moreover, the KV cache for all users requires substantial storage space, further challenging large-scale deployment.

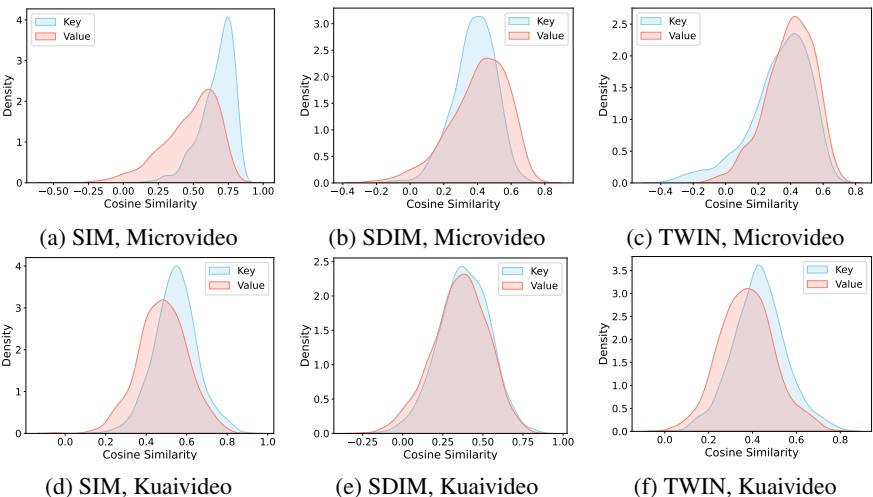

Figure 1: Probability density (estimated via kernel density estimation) of cosine similarities of K/V between different users on two datasets and three models.

These issues together highlight the necessity of developing KV cache compression techniques that can simultaneously reduce memory usage and inference latency.

## 2.2 KV CACHE COMPRESSION

Existing approaches to KV cache compression can be broadly categorized into two types: methods that reduce the sequence length $s$ of the KV cache, and methods that reduce its dimensionality $d$. In the first category, only the tokens deemed important are retained, while the KV caches of less informative tokens are discarded, along with the information they contain. For example, Loki Singhania et al. (2024) leverages PCA to compute sparse attention scores and selects the top-$k$ tokens accordingly. Similarly, Quest Tang et al. (2024) selects past tokens based on the current token, and InfiniGen Lee et al. (2024) predicts important tokens in the next layer based on the current layer's attention inputs. These methods are typically training-free but may result in performance degradation due to information loss.

The second category targets the KV dimensionality, often requiring architectural modifications and end-to-end training. A representative example is MLA Liu et al. (2024), which first projects the high-dimensional input into a low-dimensional feature space and then applies separate linear mappings to obtain K and V, thereby reducing storage requirements. While effective, these methods often introduce additional computational overhead.

Notably, these methods are primarily designed for LLMs, focusing on compressing the KV cache for individual sequences. When applied directly to recommendation scenarios, they fail to exploit the collaborative signals naturally present across users. In contrast, our work demonstrates that in sequential recommendation models, a substantial portion of KV information can be shared across users. By leveraging this cross-user similarity, the KV cache can be further compressed without sacrificing model performance.

## 3 ANALYSIS OF KEYS AND VALUES

### 3.1 CROSS-USER SIMILARITY

We first investigate whether there exists a high degree of similarity in keys and values across users, to validate the feasibility of sharing them. Specifically, taking the key as an example, we randomly sample 1,000 users from the dataset, where each user's keys are represented as matrices of shape $n \times d$ (with $n$ denoting the length of the historical sequence, which may vary across users, and $d$ denoting the attention dimension, which is consistent across users). We then compute a mean key

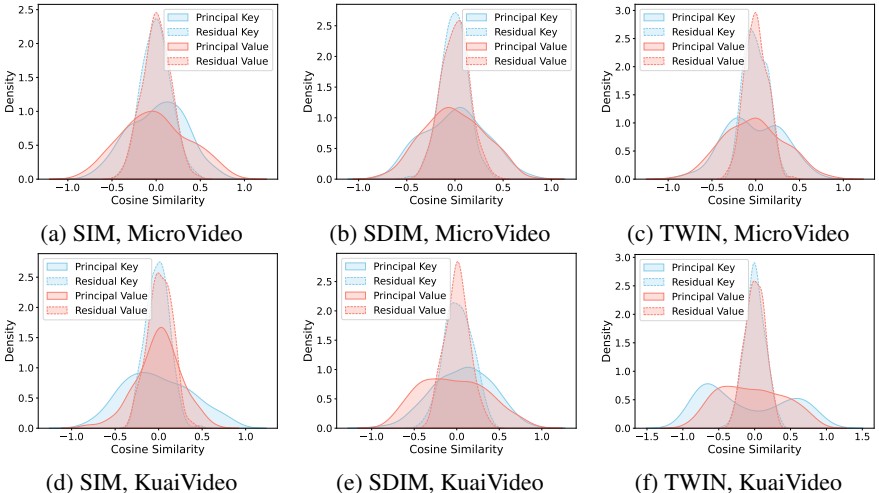

Figure 2: Probability density (estimated via kernel density estimation) of cosine similarity of principal K/V and residual K/V across users.

vector $\bar{\mathbf{K}}$ of size $1 \times d$ for each user by averaging along the sequence dimension. Therefore, the cosine similarity of the mean keys between users can be formulated as:

$$S_{\mathbf{k}}(u_i, u_j) = cosine(\bar{\mathbf{K}}_i, \bar{\mathbf{K}}_j), \quad \bar{\mathbf{K}} = \frac{1}{s} \sum_{n=1}^{s} \mathbf{K}_{n,:} \in \mathbb{R}^{1 \times d}. \tag{1}$$

Next, we randomly select the mean key vector of a user and compute its cosine similarities with the mean key vectors of all other users. Finally, we estimated the probability density of these similarities via kernel density estimation (KDE). As shown in Figure 1, on three models and two datasets, the keys and values exhibit a high degree of positive cosine similarity across users, which suggests that sharing K/V representations among users is feasible.

## 3.2 KV INFORMATION SHAREABILITY

To investigate the proportion of shareable information in the KV cache, we project the K/V into the principal component subspace and the residual subspace, and then compute their cross-user similarities. Specifically, taking the key vector $\mathbf{K} \in \mathbb{R}^{n \times d}$ (where the sequence length $n \geq d$ and the dimension $d = 256$) as an example, we first perform singular value decomposition (SVD) on it,

$$\mathbf{K} = U \Sigma V^*, \quad V \in \mathbb{R}^{d \times d}. \tag{2}$$

Then we project the key into the top-$k$ ($k = 10$ in experiments) principal components to obtain the principal component key $\mathbf{K}_p \in \mathbb{R}^{n \times k}$, which preserves the majority of the information (typically $\geq 90\%$ when $k = 10$). Subsequently, we project the key into the remaining residual dimensions to obtain the residual key $\mathbf{K}_r \in \mathbb{R}^{n \times (d-k)}$, which carries only a small fraction of the information (typically $< 10\%$ when $k = 10$),

$$\mathbf{K}_p = \mathbf{K}V[:, 1:k], \quad \mathbf{K}_r = \mathbf{K}V[:, k+1:d]. \tag{3}$$

Next, we compute the cross-user similarity based on the principal keys and residual keys,

$$S_{\mathbf{k}}^p(u_i, u_j) = cosine(\bar{\mathbf{K}}_i^p, \bar{\mathbf{K}}_j^p), \quad S_{\mathbf{k}}^r(u_i, u_j) = cosine(\bar{\mathbf{K}}_i^r, \bar{\mathbf{K}}_j^r). \tag{4}$$

Finally, we estimate their probability distributions using the method in Sec. 3.1. As shown in Figure 2, the similarity distribution of principal K/V is relatively flat, spanning a wide range between (-1, 1). In contrast, the similarity distribution of residual K/V is highly concentrated around zero, with most values falling within the range (-0.5, 0.5). Consequently, principal K/V, which carry the majority of information, are more likely to exhibit cross-user correlations, whereas residual K/V, containing much less information, are more user-specific. This leads to the conclusion that most of the information contained in K/V can potentially be shared across users.

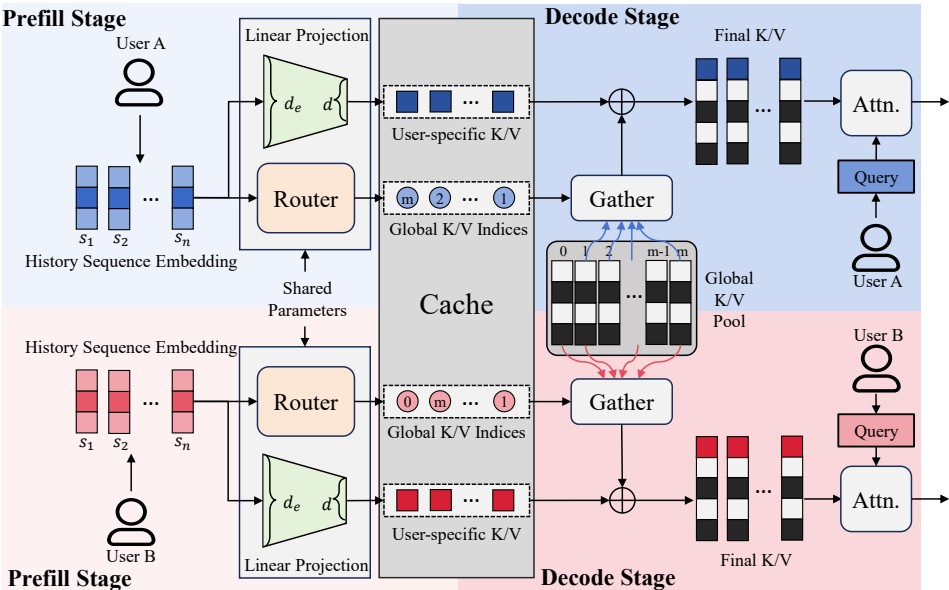

Figure 3: Overall framework of CollectiveKV. All the network modules, including the router network, the linear projection layer, and the global K/V pool, have the same parameters across different users. During prefilling, user-specific K/V and global K/V indices are calculated and cached.

## 4 COLLECTIVEKV

### 4.1 DECOMPOSING KV

As analyzed in Sec 3, the information contained in the KV can be decoupled into two parts: user-specific information and cross-user shared information. Based on this analysis, we present CollectiveKV, an item-level cross-user KV sharing strategy. In CollectiveKV, the KV is decomposed into two components: the user-specific KV and the collective KV. The user-specific KV is low-dimensional and encodes the personalized signals unique to each user's behavior sequence. In contrast, the collective KV is high-dimensional and shared across users, capturing the collaborative information that is common in recommendation scenarios.

### 4.2 FRAMEWORK

In real-time sequential recommendation systems, the computation of the attention module can be divided into two stages—the prefill stage and the decode stage—to reduce inference latency. In the prefill stage, the user's historical sequence is projected into keys and values and stored as a KV cache. When a user request arrives, the user enters the decode stage, where the cached KV is retrieved to compute attention scores with the user's query.

As shown in Figure 3, our method follows the practice of separating the prefill and decode stages.

**In the prefill stage,** given an input embedding $\mathbf{S} = \{s_1, s_2, ..., s_n\} \in \mathbb{R}^{n \times d_e}$ (typically the user history sequence embedding) with length of $n$ and dimension of $d_e$, we use a linear projection to reduce its dimensionality and get user-specific key $\mathbf{K}_u \in \mathbb{R}^{n \times d_u}$ or value $\mathbf{V}_u \in \mathbb{R}^{n \times d_u}$, denoted as,

$$\mathbf{K}_u = \mathbf{S}W_{\mathbf{k}} + b_{\mathbf{k}}, \quad \mathbf{V}_u = \mathbf{S}W_{\mathbf{v}} + b_{\mathbf{v}}, \tag{5}$$

where $W_{\mathbf{k}}, W_{\mathbf{v}} \in \mathbb{R}^{d_e \times d_u}$ are learnable projection matrices and $b_{\mathbf{k}}, b_{\mathbf{v}} \in \mathbb{R}^{d_u}$ are learnable bias.

Meanwhile, a router network maps the sequence embedding into a series of item-level global K/V indices $\mathbf{I}_{\mathbf{k}}, \mathbf{I}_{\mathbf{v}} \in \mathbb{N}^{n \times 1}$. The details of the router are presented in the next section. After that, both the global K/V indices and the user-specific K/V will be cached.

**In the decode stage,** both of them will be retrieved from the cache. The indices serve to gather item-level keys or values from a learnable global K/V pool $P_{\mathbf{k}}, P_{\mathbf{v}} \in \mathbb{R}^{m \times d_g}$ to form the collective

$\mathbf{K}_c, \mathbf{V}_c \in \mathbb{R}^{n \times d_g}$, where $m$ is the pool size and $d_g$ is the global K/V dimension,

$$\mathbf{K}_c[i] = P_\mathbf{k}[\mathbf{I_k}[i]], \quad \mathbf{V}_c[i] = P_\mathbf{v}[\mathbf{I_v}[i]], \quad i = 1, ..., n. \tag{6}$$

Then, we concatenate user-specific K/V and collective K/V to get the final $\mathbf{K}, \mathbf{V} \in \mathbb{R}^{n \times d_a}$ ($d_a = d_u + d_g$) as,

$$\mathbf{K} = concat(\mathbf{K}_u, \mathbf{K}_c), \mathbf{V} = concat(\mathbf{V}_u, \mathbf{V}_c). \tag{7}$$

Finally, the final KV and the user's query serve to compute the attention output.

## 4.3 COLLECTIVEKV ROUTER

The information across users is shared through the global K/V pool with a special router network, which is end-to-end trained. As illustrated in Figure 3, the router maps the embedding into global K/V indices, which are then used to gather the collective K/V from the pool.

Specifically, taking the key as an example, the router first projects the embedding $\mathbf{S} \in \mathbb{R}^{n \times d_e}$ into a routing map $\mathbf{M} \in \mathbb{R}^{n \times m}$. Then, for each item, it selects the index of the maximum element along the second dimension to form the global key indices $\mathbf{I}_k \in \mathbb{N}^{n \times 1}$:

$$\mathbf{M} = \mathbf{S}W_r + b_r, \quad \mathbf{I_k}[i] = \arg\max_j \mathbf{M}_{ij}, \tag{8}$$

where $W_r, b_r$ are parameters to be learned. After that, the collective keys are gathered according to Eq. 6. In the next step, the model behaves slightly differently during inference and training.

**During inference**, it directly concatenates the collective keys and the user-specific keys as Eq. 7. However, if the same operation is applied during training, the gradients cannot be propagated through the indices back to the router network. Therefore, **during training**, we multiply the collective key by the corresponding sigmoid output of the router, through which gradients can be successfully backpropagated to the router. So the Eq. 6 is actually formulated as:

$$\mathbf{K}_c[i] = \begin{cases} \sigma\big(\mathbf{M}[i, \mathbf{I_k}[i]]\big) \cdot P_\mathbf{k}[\mathbf{I_k}[i]], & \text{if training,} \\ P_\mathbf{k}[\mathbf{I_k}[i]], & \text{if inference,} \end{cases} \quad i = 1, \dots, n, \tag{9}$$

where $\sigma$ denotes the sigmoid activation function. However, we need to ensure consistency between training and inference by encouraging the sigmoid weights to be close to 1. At this point, the reason we chose sigmoid activation becomes clear: simply encouraging the input weights to be large could guarantee the sigmoid outputs close to 1. Consequently, we introduce a peak loss to supervise the outputs of the router network:

$$\mathcal{L}_{\text{peak}} = -\frac{1}{n} \sum_{i=1}^{n} \log\big(\sigma(\mathbf{M}[i, \mathbf{I_k}[i]])\big). \tag{10}$$

Meanwhile, to ensure that each key in the global key pool learns meaningful information, we need to encourage every key to have an equal probability of being selected. So we introduce a load balance loss to supervise the outputs of the router as:

$$\mathcal{L}_{\text{balance}} = \sum_{j=1}^{m} \bar{p}_j \log \frac{\bar{p}_j}{u}, \quad \bar{p}_j = \frac{1}{n} \sum_{i=1}^{n} \text{softmax}(\mathbf{M})_{i,j}, \quad u = \frac{1}{m}, \tag{11}$$

where $\bar{p}_j$ denotes the average probability of selecting the $j$-th key across the sequence, and $u$ is the uniform probability. This loss is the KL divergence between the average selection probabilities of the keys and a uniform distribution.

## 4.4 LATENCY ANALYSIS

During the decode stage, the latency introduced by the original KV cache technique mainly stems from the loading delay, i.e., transferring K/V from external storage to GPU memory. This latency can be described as:

$$T_{load}(sd_a) = T_{setup} + \frac{sd_a}{B}, \tag{12}$$

Table 1: The performance of our method on different models and datasets. The CR metric denotes the compression rate of the KV cache. Ebnerd Kruse et al. (2024) has not been adapted for HSTU in our work; thus, we do not include it for comparison in the table.

| | MicroVideo | | | | KuaiVideo | | | | Ebnerd | | | |
|---|---|---|---|---|---|---|---|---|---|---|---|---|
| | GAUC↑ | AUC↑ | Logloss↓ | CR↓ | GAUC↑ | AUC↑ | Logloss↓ | CR↓ | GAUC↑ | AUC↑ | Logloss↓ | CR↓ |
| SIM | 0.6954 | 0.6933 | 0.4282 | / | 0.6577 | 0.6798 | 0.4620 | / | 0.6975 | 0.7024 | 0.2754 | / |
| SIM+ours | **0.6973** | **0.7057** | **0.4203** | **0.016** | **0.6604** | **0.6900** | **0.4520** | **0.012** | **0.7003** | **0.7088** | **0.2749** | **0.027** |
| SDIM | 0.6857 | 0.6749 | 0.4389 | / | 0.6506 | 0.6766 | 0.4611 | 0 | 0.6890 | 0.6948 | 0.2760 | / |
| SDIM+ours | **0.6883** | **0.6871** | **0.4323** | **0.012** | **0.6545** | **0.6786** | **0.4583** | **0.016** | **0.6941** | **0.6956** | **0.2713** | **0.008** |
| ETA | 0.6960 | 0.6911 | 0.4325 | / | **0.6699** | 0.6886 | 0.4577 | / | 0.6931 | 0.6990 | 0.2717 | / |
| ETA+ours | **0.6984** | **0.7015** | **0.4295** | **0.012** | 0.6677 | **0.6889** | **0.4561** | **0.020** | **0.7020** | **0.7032** | **0.2686** | **0.027** |
| TWIN | 0.6984 | 0.7064 | 0.4233 | 0 | 0.6483 | 0.6756 | **0.4618** | / | 0.6969 | 0.7028 | 0.2710 | / |
| TWIN+ours | **0.6990** | **0.7074** | **0.4226** | **0.016** | **0.6558** | **0.6786** | 0.4627 | **0.020** | **0.7051** | **0.7081** | **0.2679** | **0.016** |
| HSTU | 0.6859 | 0.7381 | 0.4101 | / | 0.6663 | 0.7454 | **0.4388** | / | - | - | - | - |
| HSTU+ours | **0.6862** | **0.7385** | **0.4100** | **0.043** | **0.6668** | **0.7475** | 0.4398 | **0.012** | - | - | - | - |

where $sd_a$ is the KV cache size, $T_{setup}$ is the transfer setup latency, and $B$ is the bus bandwidth.

In our approach, the loading latency is denoted as $T_{load}(sd_u + s)$, where $d_u \ll d_a$ and the external $s$ represents the global KV indices. After this loading step, the model needs to gather the collective KV from the global K/V pool according to the global KV indices. Since the global K/V pool resides entirely in GPU memory, this operation amounts to a GPU memory indexing, whose latency is much lower compared to $T_{load}$. We denote the latency of this indexing operation as $T_{index}(sd_g)$, where $sd_g$ is the collective KV size.

In summary, the latency of our method is $T_{load}(sd_u + s) + T_{index}(sd_c)$. The latency comparison experiments with the KV cache are presented in Sec. 5.2.

# 5 EXPERIMENTS

## 5.1 MODELS, DATASETS AND METRICS

**Models.** To comprehensively evaluate the effectiveness of our proposed KV sharing strategy, we consider both target-attention-based and self-attention-based recommendation models. Specifically, we select four representative models built upon the target attention paradigm: SIM Pi et al. (2020), SDIM Cao et al. (2022), ETA Chen et al. (2022), and TWIN Chang et al. (2023). In addition, we reproduce HSTU Zhai et al. (2024), a self-attention-based model that has recently introduced the KV cache mechanism into recommendation. The implementations of the above baseline models and the corresponding experimental pipeline largely reuse the existing FuxiCTR codebase and experimental setups from prior work Zhu et al. (2022; 2021).

**Datasets.** We evaluate our method on three long-sequence CTR prediction datasets. Specifically, we use MicroVideo Chen et al. (2018), a widely adopted benchmark dataset for short-video recommendation; KuaiVideo, which originates from the Kuaishou competition at ChinaMM 2018 and contains rich user–video interaction sequences; and EBNeRD-Small Kruse et al. (2024), a Danish news recommendation dataset. These datasets are characterized by long user behavior histories, which make them suitable for evaluating the efficiency and effectiveness of KV caching strategies. Moreover, the dataset splitting strategy and configuration settings follow the protocol introduced in LAIN Zhang et al. (2026).

**Metrics.** We adopt three widely used evaluation metrics: AUC, GAUC, and Logloss. In addition, we introduce the Compression Rate (CR) to assess the efficiency of KV cache compression, defined as the ratio between the reduced KV cache size and the original one. Note that the CollectiveKV cache includes two parts: the global KV indices and the user-specific KV.

## 5.2 COMPARISON WITH BASELINE

The baseline setting is keeping the model unchanged, *i.e.*, retaining the original target attention or self-attention structure. As shown in Table 1, our proposed method significantly reduces the KV

Table 2: Latency analysis in comparison with KV cache, conducted on an A100 GPU.

| Batch Size | 1 | 8 | 32 | 64 | 128 | 256 | 512 |
|---|---|---|---|---|---|---|---|
| KV Cache Latency (ms) | 0.099 | 1.030 | 1.808 | 3.418 | 6.679 | 15.216 | 32.991 |
| CollectiveKV Latency (ms) | 0.084 | 0.129 | 0.136 | 0.152 | 0.222 | 0.375 | 0.695 |

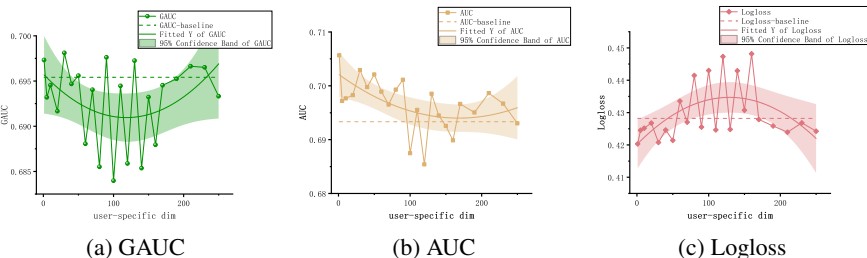

(a) GAUC      (b) AUC      (c) Logloss

Figure 4: The influence of the user-specific feature dimension on the model performance.

cache size, achieving a compression ratio as low as 0.008, while maintaining or even enhancing model performance compared to the baseline.

Specifically, in two target attention models, SIM Pi et al. (2020) and SDIM Cao et al. (2022), our method not only substantially compresses the KV cache but also improves model performance. In the other two target attention models, ETA Chen et al. (2022) and TWIN Chang et al. (2023), our method also achieves better performance on the MicroVideo and Ebnerd datasets. In the self-attention model HSTU Zhai et al. (2024), our method also surpasses the baseline on the MicroVideo dataset. On the Kuaivideo dataset, our method also achieves higher GAUC and AUC with a compression rate of 0.012. These satisfying results demonstrate that CollectiveKV can effectively capture collaborative information among users, thereby enhancing recommendation accuracy.

Furthermore, we conduct experiments to measure and compare the actual latency between the standard KV cache and our CollectiveKV cache. The experiments are conducted on the SIM model and the Kuaivideo dataset. For the standard KV cache, we measure the average time required to load a batch of keys from CPU memory to GPU memory as its latency. For CollectiveKV, we measure the total time in the decode stage before the attention calculation as its latency.

As shown in Table 2, our method achieves significantly lower latency compared to the original KV cache. Moreover, as the batch size increases, the latency of the KV cache grows sharply. Specifically, when the batch size increases from 1 to 512, the latency of the KV cache rises from 0.099 ms to 32.991 ms; in contrast, the latency of our method only increases from 0.084 ms to 0.695 ms.

## 5.3 INFLUENCE OF THE USER-SPECIFIC INFORMATION

As analyzed in Sec 3.2, user-specific information might constitute only a small fraction of the overall K/V representations, while the majority of the information could be shared across users. Here, we investigate how varying the proportion of user-specific information affects the model performance.

Specifically, we evaluate model performance under different user-specific KV dimensions using the SIM model and the MicroVideo dataset, and fit the results with a quadratic polynomial. As shown in Figure 4, the model performance first decreases and then increases. This is consistent with the findings in Section 3.2: a smaller user-specific dimension better matches the distribution of collaborative and user-specific signals. However, as the dimension continues to increase, the shared dimension decreases, and CollectiveKV gradually degenerates into the original KV.

## 5.4 DISCUSSION OF THE GLOBAL K/V POOL SIZE

The size of the global K/V pool determines the upper bound of the capacity for learnable shared information. As the user population grows, the amount of shared information might increase, necessitating a corresponding enlargement of the global K/V pool. To investigate how the pool size

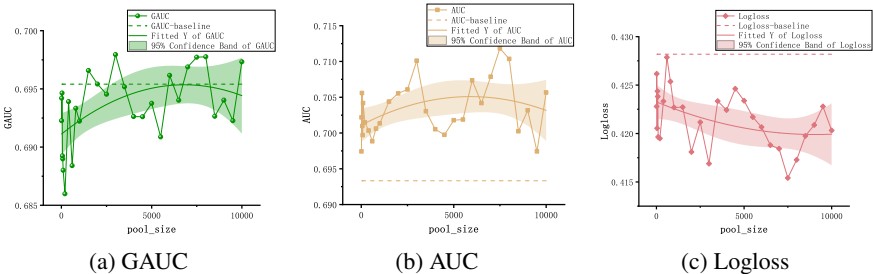

Figure 5: The influence of the global K/V pool size on the model performance.

Table 3: Ablation experiments on the peak loss and the balance loss.

| | MicroVideo | | | KuaiVideo | | | Ebnerd | | |
| --- | --- | --- | --- | --- | --- | --- | --- | --- | --- |
| | GAUC | AUC | Logloss | GAUC | AUC | Logloss | GAUC | AUC | Logloss |
| Baseline | 0.6954 | 0.6933 | 0.4282 | 0.6577 | 0.6798 | 0.4620 | 0.6975 | 0.7024 | 0.2754 |
| CollectiveKV | 0.6971 | **0.7105** | 0.4339 | **0.6604** | **0.6900** | **0.4520** | 0.6976 | 0.7011 | **0.2727** |
| +peak loss | 0.6964 | 0.7058 | 0.4221 | 0.6599 | 0.6892 | 0.4531 | 0.6964 | 0.7017 | 0.2772 |
| +balance loss | 0.6941 | 0.7038 | 0.4999 | 0.6588 | 0.6856 | 0.4554 | 0.6967 | 0.6995 | 0.2729 |
| +all losses | **0.6973** | 0.7057 | **0.4203** | 0.6585 | 0.6890 | 0.4533 | **0.7003** | **0.7088** | 0.2749 |

influences model performance, we conduct a series of experiments with different pool sizes on the SIM model and the MicroVideo dataset.

As shown in Figure 5, the polynomial fitting results reveal that GAUC and AUC metrics exhibit upward trends as the pool size increases to around 7000, after which both metrics begin to decline. The Logloss metric exhibits the opposite trend. Therefore, we conjecture that enlarging the pool beyond the capacity needed to capture shared information may instead expand the representation space unnecessarily, introducing noise and reducing the router's routing accuracy, which in turn can lead to performance degradation.

## 5.5 ABLATION STUDY

We conduct ablation experiments to verify the effectiveness of the key components in our method. First, we examine the impact of the proposed peak loss $\mathcal{L}_{peak}$ and load balance loss $\mathcal{L}_{balance}$ on training the router network. As shown in Table 3, our method achieves the best performance on the MicroVideo and Ebnerd datasets when both losses are applied, which demonstrates their effectiveness. On the KuaiVideo dataset, due to its substantial differences from the other two datasets—such as longer average sequence lengths and more challenging prediction tasks—the two losses perform suboptimally under the same loss weight settings. Nonetheless, the results with both losses still outperform the baseline, which further demonstrates the advantage of our method.

Furthermore, we analyze the impact of sharing keys and values separately on the model performance, as shown in Table 4. On the KuaiVideo and Ebnerd datasets, the model achieves the best performance when both keys and values are shared. On the MicroVideo dataset, sharing only the keys (CollectiveK) yields the highest GAUC, while sharing only the values (CollectiveV) results in the best AUC and Logloss. Meanwhile, sharing both K and V simultaneously achieves the second-best performance across all three metrics. These results suggest that the optimal sharing strategy may vary across datasets, indicating a trade-off between key and value sharing.

## 6 LIMITATIONS AND DISCUSSION

A key limitation of our method lies in determining the appropriate dimensionality for the user-specific KV. The choice of this dimension directly impacts the overall compression rate: if it is set too high, the KV cache will remain large and model performance may decrease; if set too low, the model may fail to capture sufficient user-specific information, potentially harming performance. Currently, based on empirical observations, the user-specific KV dimension can be set very small,

Table 4: Ablation experiments on sharing keys or values.

| | MicroVideo | | | KuaiVideo | | | Ebnerd | | |
|---|---|---|---|---|---|---|---|---|---|
| | GAUC | AUC | Logloss | GAUC | AUC | Logloss | GAUC | AUC | Logloss |
| Baseline | 0.6954 | 0.6933 | 0.4282 | 0.6577 | 0.6798 | 0.4620 | 0.6975 | 0.7024 | 0.2754 |
| CollectiveK | **0.6977** | 0.6936 | 0.4281 | **0.6627** | 0.6878 | 0.4534 | 0.6947 | 0.6991 | 0.2820 |
| CollectiveV | 0.6965 | **0.7073** | **0.4179** | 0.6591 | 0.6872 | 0.4538 | 0.6974 | 0.6998 | 0.2780 |
| CollectiveKV | 0.6973 | 0.7057 | 0.4203 | 0.6604 | **0.6900** | **0.4520** | **0.7003** | **0.7088** | **0.2749** |

typically within 4% of the attention head dimension, while still achieving optimal performance. Developing an automated or adaptive strategy to determine the optimal user-specific KV dimension remains an important direction for future work.

Overall, our work makes two main contributions. First, we introduce a new paradigm for KV cache compression by sharing KV information across users. This paradigm could potentially be extended to other application scenarios beyond sequential recommendation. Second, we propose a novel KV cache sharing mechanism that significantly compresses the KV cache and reduces inference latency, while simultaneously maintaining or improving model performance.

ACKNOWLEDGMENTS

This work was supported by the National Natural Science Foundation of China (62302532), Guangdong Basic and Applied Basic Research Foundation (2025A1515011224), and Shenzhen Science and Technology Program (KQTD20221101093559018, SYSRD20250529113401002).

Furthermore, we would like to thank the contributors of the baseline codebase Zhu et al. (2022; 2021); Zhang et al. (2026) used in this work. The provided implementations and experiment setups offered important support for reproducing the baseline models and conducting the empirical studies in this paper.

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

## A  LLM Usage Statement

Large Language Models (LLMs) were solely employed for improving the writing clarity and language fluency of this paper, and were not involved in the idea generation, methodology design, experiments, or analysis.

## B  Code and Other Supplementary Materials

We provide the code of our method in the repository: `https://github.com/Lee-Jingyu/CollectiveKV`

## C  Implementation Details

In our experiments, we set the global K/V pool size $m$ to 10000 by default. The weights of the peak loss $\mathcal{L}_{\text{peak}}$ and the load balance loss $\mathcal{L}_{\text{balance}}$ are 0.01 and 1.0, respectively. The target-attention-based models are from the LongCTR benchmark: `https://github.com/reczoo/LongCTR`. The experiments are conducted on a single A100 GPU, with Intel(R) Xeon(R) Platinum 8358 CPU @ 2.60GHz, and 1TB RAM, based on Pytorch v2.0.1.

## D  Visualization of Router Activations

To gain deeper insights into the behavior of the router network, we provide a visualization of its activation patterns. Specifically, we randomly select 100 users and collect the indices of the activated keys produced by the router outputs. We then plot the histogram of the activation indices with a bin size of 50, as shown in Figure 6.

From the histograms, we observe that the router activations are distributed across different indices rather than being concentrated on some special keys, which demonstrates the effectiveness of the peak loss and balance loss in guiding the router network. This balanced activation pattern indicates that the router is able to leverage diverse components of the global KV pool, thereby enhancing the model's ability to capture shared information while avoiding expert collapse.

## E  Analysis of Router Activations for Highly Similar Users

To further investigate the relationship between user similarity and router activations, we select two users whose history sequence embeddings exhibit a cosine similarity higher than 0.8. We then plot the histograms of their router activation indices, using a bin size of 100, as shown in Figure 7.

From the histograms, it is evident that the two users share a large number of activation indices, reflecting the high similarity in their behavior sequences. Nevertheless, there are also a few indices that are not shared, indicating user-specific differences captured by the router. Quantitatively, the overlap ratio of activated indices between the two users, whose similarity is 0.9887, is 0.8571.

This analysis demonstrates that the router effectively balances shared and user-specific information: highly similar users tend to activate largely overlapping portions of the global KV pool, while still maintaining a small amount of unique activations to preserve individual preferences.

## F  Discussion of the sharing mechanism

We believe that our method outperforms the baseline because it explicitly decouples the collaborative signals and the user-specific signals in the feature space. In the original K/V representations, these two types of signals share the same feature space of dimension $d_a$. This inevitably leads to a certain degree of information entanglement, making it difficult for the model to distinguish and effectively utilize the two components.

In contrast, CollectiveKV assigns the collaborative and user-specific signals to separate subspaces: the first $d_u$ dimensions of the K/V vectors encode user-specific information, while the remaining

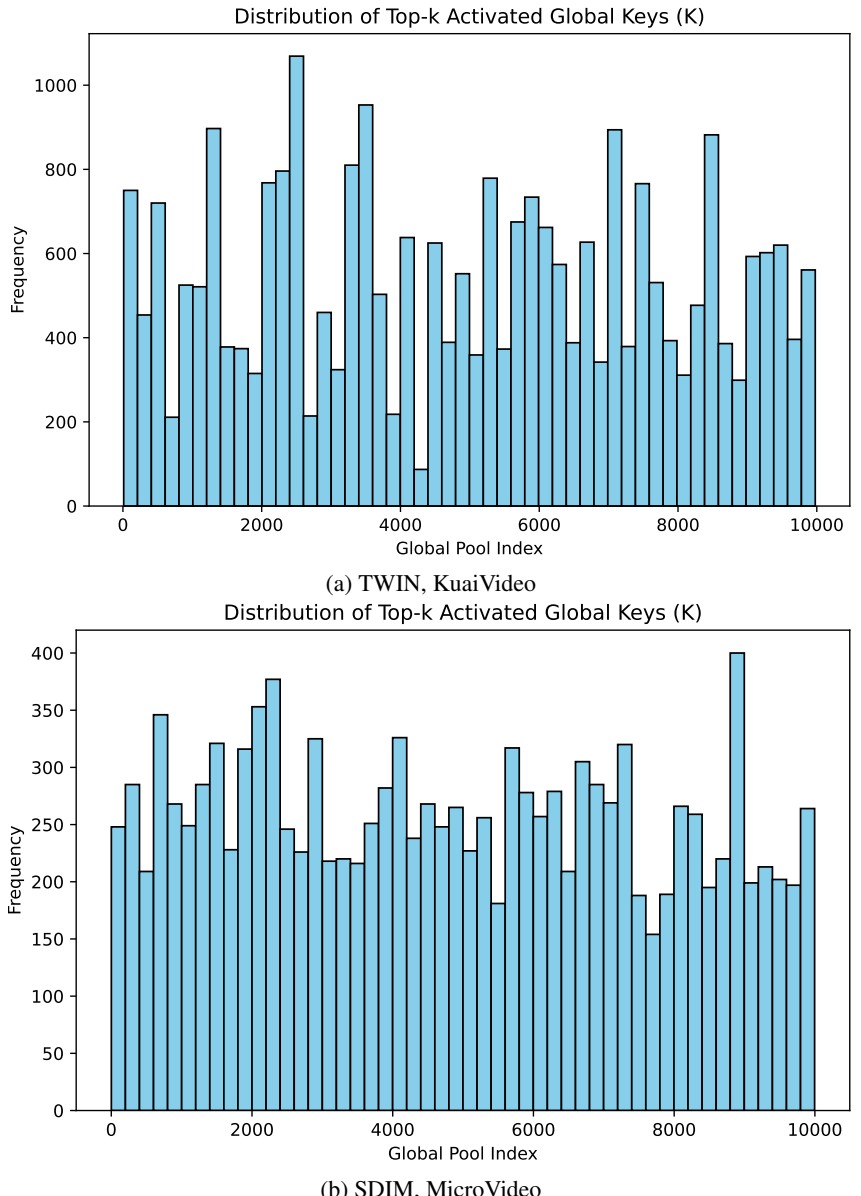

(a) TWIN, KuaiVideo

(b) SDIM, MicroVideo

Figure 6: Visualization of Router Activations.

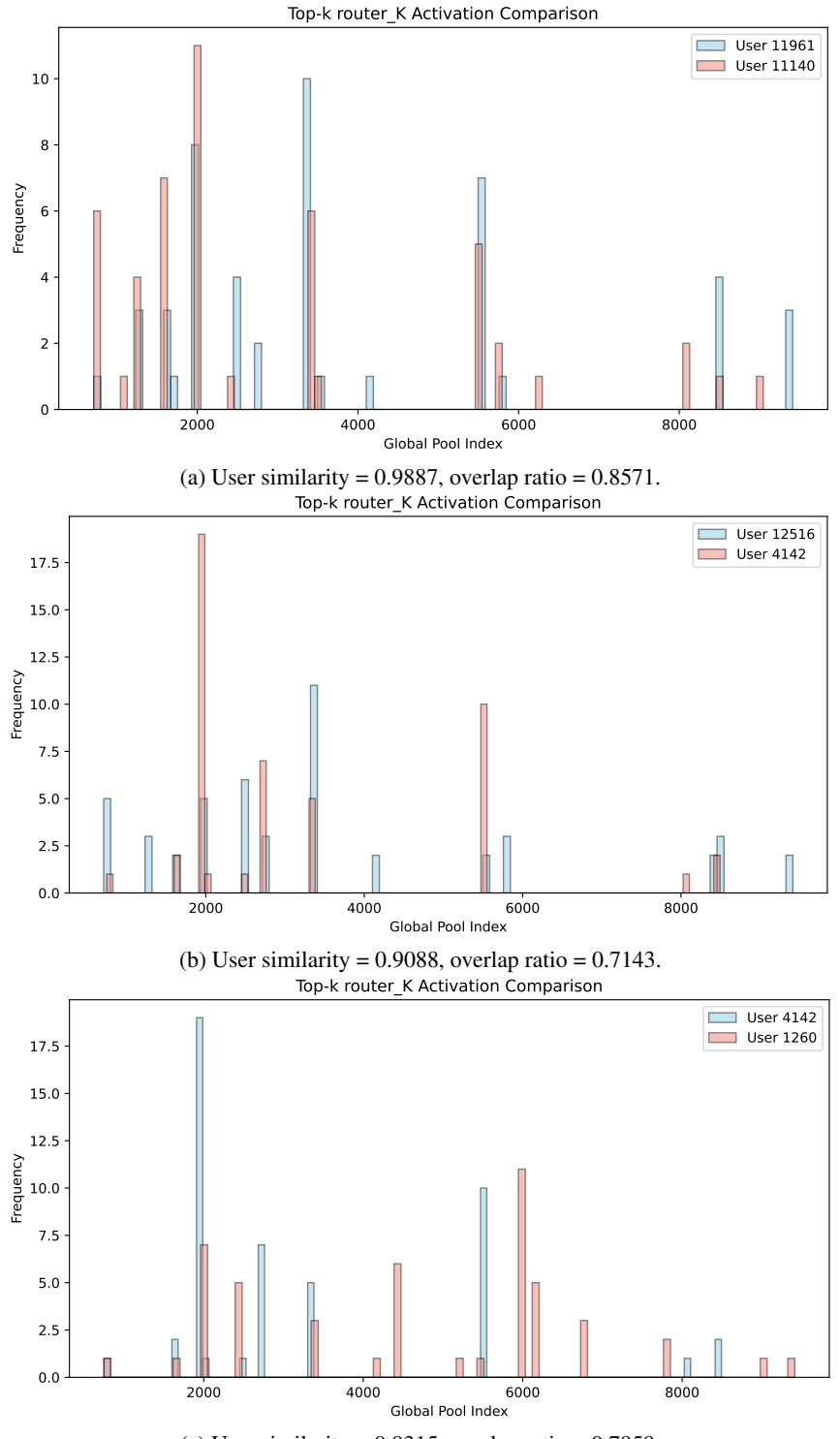

(a) User similarity = 0.9887, overlap ratio = 0.8571.

(b) User similarity = 0.9088, overlap ratio = 0.7143.

(c) User similarity = 0.8315, overlap ratio = 0.7959.

Figure 7: Router activation histograms for two users with high cosine similarity in their history sequence embeddings. Bin size is 100.

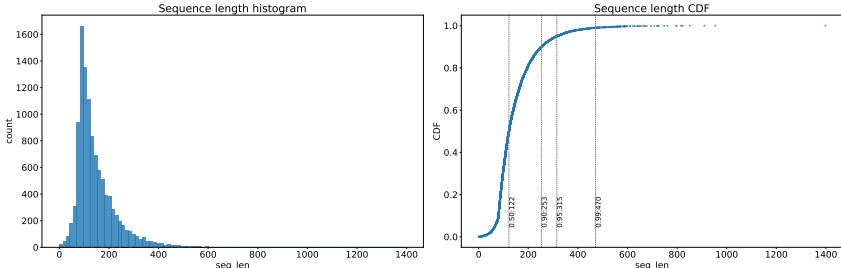

Figure 8: Sequence length distribution of the MicroVideo dataset.

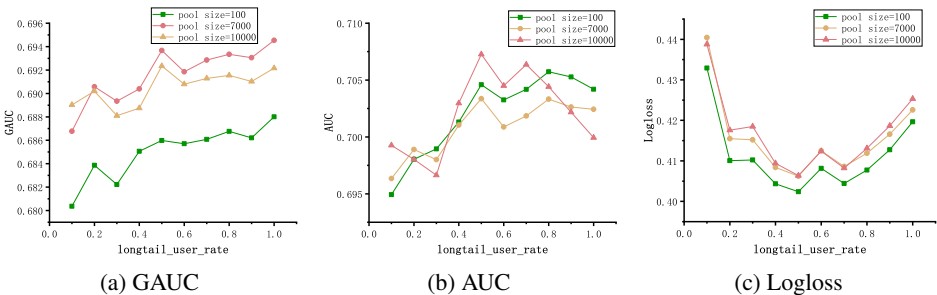

Figure 9: Model performance on long-tail users with different global KV pool sizes.

$d_g$ dimensions encode collaborative information, with $d_u + d_g = d_a$. Under the constraint of a fixed total dimensionality, this explicit separation enables a clearer decomposition of the two types of signals, making it easier for the model to identify and leverage them effectively.

A similar theoretical perspective comes from domain-invariant disentangled representation learning: if each user is regarded as a domain, then we can decompose the K/V representations into a domain-invariant (shared) component and a domain-specific (user-specific) residual. Another similar scenario is multi-domain recommendation, in which users are clustered into different domains according to some specific features. In our work, we can treat each user as a special domain, since each user has a unique behavior pattern. Considering different user group has within-group collaborative signal, these signals can be compressed into domain invariant parameters (shared pool). And their specific behaviors can be modeled as domain specific parameters.

## G    INFLUENCE OF THE GLOBAL KV POOL CAPACITY ON LONG-TAIL USERS

In recommender systems, long-tail users refer to the large group of users with very short behavior sequences, who typically account for a high proportion of the entire user base. Because their sequences are short, it is usually difficult to generate accurate recommendations for these users. Moreover, the collaborative signals contained in such short sequences are harder to capture.

Therefore, we aim to investigate whether a global KV pool with insufficient capacity may struggle to learn the behavioral patterns of long-tail users. First, we analyze the distribution of user behavior sequence lengths in the MicroVideo dataset. As shown in Figure 8, 50% of users have sequence lengths shorter than 122, 90% shorter than 253, and fewer than 1% have lengths greater than 470. This indicates that the long-tail user issue also exists in this dataset.

We then partition the dataset based on user sequence lengths, isolating the long-tail users with the shortest sequences and evaluating the model specifically on them. In the experiments, we set a very small KV pool size of 100 to ensure that the pool capacity is insufficient to capture all collaborative signals in the dataset. We conduct multiple experiments by varying the parameter *longtail user rate*, which determines the proportion of users included in the evaluation. For example, when *longtail user rate* = 0.1, we test on only 10% of users with the shortest behavior sequences; when *longtail user rate* = 1.0, all users are tested.

The experimental results are shown in Figure 9. We observe that long-tail users exhibit lower GAUC and AUC, as well as higher logloss, no matter what pool size is chosen. These results indicates that long-tail users typically have weaker collaborative signals, making it harder to model their behavior pattern.

