# OpenReview forum: "CollectiveKV: Decoupling and Sharing Collaborative Information in Sequential Recommendation"
_ICLR.cc/2026/Conference — ICLR 2026 Poster_

### Official Review · Reviewer_peMb · 2025-10-30

**Soundness:** 2
**Presentation:** 3
**Contribution:** 2
**Rating:** 4
**Confidence:** 4

**Summary:**

This paper proposes CollectiveKV, a cross-user KV cache sharing mechanism for sequential recommendation models. The key idea is to decouple the KV cache into a high-dimensional global KV shared across users and a low-dimensional user-specific KV. CollectiveKV employs a router network to retrieve shared KV items from a global pool and concatenates them with user-specific KV for attention computation. Experimental results on five sequential recommendation models and three datasets show large cache compression rates while reportedly maintaining or even improving accuracy metrics compared to baselines.

**Strengths:**

- Addresses a relevant efficiency problem in sequential recommendation — KV cache storage and latency — with a novel sharing-based compression approach.
- The method design is conceptually simple yet elegant, with clear separation of shared and user-specific KV components.
- Experimental evaluation spans multiple models and datasets, reporting consistent compression gains and performance improvements.
- The work offers practical value for large-scale industrial recommendation systems where KV cache size is a bottleneck.

**Weaknesses:**

- While the work is innovative, the overall innovation point is relatively simple and may not be sufficient for a strong theoretical contribution. It is more like a balance between the existing group clustering-based attention and the ordinary attention.
- There is a theoretical concern: since global modeling is used to compress the KV cache, the actual recommendation performance should, in principle, be at most the same as before compression. However, the reported experiments show that the proposed method is significantly superior to the baseline, which raises questions about the cause. Further exploration experiments and deeper analysis are needed to explain why and under what conditions the method can outperform the standard uncompressed KV cache.

**Questions:**

- The novelty is effective but relatively minor. It is more like a balance between the existing group clustering-based attention and the ordinary attention.
- Further exploration experiments and deeper analysis are needed to explain why and under what conditions the method can outperform the standard uncompressed KV cache.

---

> ### Author Response · Authors · 2025-11-22
>
> W1/Q1. While the work is innovative, the overall innovation point is relatively simple and may not be sufficient for a strong theoretical contribution. It is more like a balance between the existing group clustering-based attention and the ordinary attention.
>
> R1. Thank you for pointing out this concern.
> However, we are not fully certain about the “group clustering-based attention” mentioned in the review, as this term is not commonly used in the sequential recommendation literature. We wonder whether the reviewer is referring to Grouped Query Attention (GQA) or its variants in Transformer models.
>
> If so, we would like to clarify that our method is fundamentally different. GQA clusters attention heads within a batch to reduce the number of key/value projections, while our method shares K/V representations across users. The two operate at different levels. In addition, GQA assumes similarity between attention heads in the same model layer. CollectiveKV assumes collaborative signals among users, supported by our empirical SVD and similarity analyses (Sec. 3).
>
> W2/Q2. Theoretical concern
>
> R2. We thank the reviewer for raising this important theoretical concern.
> We believe that our method outperforms the baseline because it explicitly decouples the collaborative signals and the user-specific signals in the feature space. In the original K/V representations, these two types of signals share the same feature space of dimension $d_a$. This inevitably leads to a certain degree of information entanglement, making it difficult for the model to distinguish and effectively utilize the two components.
>
> In contrast, CollectiveKV assigns the collaborative and user-specific signals to separate subspaces: the first $d_u$ dimensions of the K/V vectors encode user-specific information, while the remaining $d_g$ dimensions encode collaborative information, with $d_u+d_g=d_a$. Under the constraint of a fixed total dimensionality, this explicit separation enables a clearer decomposition of the two types of signals, making it easier for the model to identify and leverage them effectively.
>
> A similar theoretical perspective comes from domain-invariant disentangled representation learning: if each user is regarded as a domain, then we can decompose the K/V representations into a domain-invariant (shared) component and a domain-specific (user-specific) residual. Another similar scenario is multi-domain recommendation, in which users are clustered into different domains according to some specific features. In our work, we can treat each user as a special domain, since each user has a unique behavior pattern. Considering different user group has within-group collaborative signal, these signals can be compressed into domain invariant parameters (shared pool). And their specific behaviors can be modeled as domain specific parameters.

---

### Official Review · Reviewer_uWXd · 2025-11-01

**Soundness:** 3
**Presentation:** 3
**Contribution:** 3
**Rating:** 6
**Confidence:** 4

**Summary:**

This paper, "CollectiveKV," addresses the significant challenge of storage overhead and inference latency introduced by the Keys and Values (K/V) cache mechanism in sequential recommendation systems, particularly those leveraging Transformer attention. Sequential recommendation models, unlike LLMs, serve a massive user base with potentially long interaction histories, leading to KV cache sizes that exceed GPU capacity and necessitate costly offloading.

The core insight driving this work is the observation that K/V sequences across different users exhibit substantial similarities, suggesting the presence of collaborative signals embedded within the K/V representations. Through singular value decomposition (SVD) analysis, the authors confirm that the majority of K/V information is shareable (principal component subspace), while only a small fraction is user-specific (residual subspace).

Motivated by this, CollectiveKV proposes a novel compression paradigm that decouples the K/V into a high-dimensional collective KV shared via a learnable global pool, and a low-dimensional user-specific KV. This decomposition and sharing mechanism, governed by a router network trained with specific losses (peak and load balance loss), allows for drastic compression. The experimental results demonstrate that CollectiveKV achieves remarkable compression and latency reduction while maintaining or improving recommendation accuracy across five models and three datasets.

**Strengths:**

1. Groundbreaking Compression Rate and Performance Preservation: The method achieves an exceptional level of compression, reducing the KV cache size to as low as 0.8% of its original size (a compression ratio of 0.008). Crucially, this compression is achieved while simultaneously maintaining or even enhancing model performance across various target-attention and self-attention models (SIM, SDIM, ETA, TWIN, HSTU) and three long-sequence datasets.

2. Significant Inference Latency Reduction: CollectiveKV effectively addresses the stringent latency requirements of real-time sequential recommendation. Comparative experiments show that the proposed method yields significantly lower latency than the standard KV cache, particularly for larger batch sizes. For instance, at a batch size of 512, the standard KV cache latency was 32.991 ms, whereas CollectiveKV achieved 0.695 ms. This result confirms the practical deployment benefit of the approach in mitigating transfer latency caused by offloading large caches.

3. Novel and Well-Motivated Compression Paradigm: The paper introduces a new compression paradigm for K/V caches by sharing information across users, a unique approach that exploits the strong collaborative signals inherent in recommendation scenarios. This is a fundamental distinction from existing methods primarily designed for LLMs, which focus on compressing individual sequences

**Weaknesses:**

1. Empirical Determination of User-Specific KV Dimensionality: A primary limitation acknowledged by the authors is the difficulty in determining the optimal dimensionality ($d_u$) for the low-dimensional user-specific KV. The choice of this dimension critically affects both the overall compression rate and the model's ability to capture sufficient personalized signals; if set incorrectly, performance may degrade or compression benefits may be lost.

2. Lack of Adaptive Strategy for $d_u$: Related to the above, the current optimal setting for the user-specific KV dimension relies on empirical observations (e.g., typically set within 4% of the attention head dimension). The paper highlights that developing an automated or adaptive strategy to determine this dimension remains a crucial direction for future work, indicating that the current method relies on a tuning process for optimal results.

3. Inconsistent Robustness of Training Losses: While the peak loss and load balance loss are generally effective, the ablation study shows that they perform suboptimally on the KuaiVideo dataset under the standardized loss weight settings. This suggests that the hyperparameter weights for these auxiliary losses might need substantial dataset-specific tuning to maintain optimal performance across diverse recommendation environments.

4. Dataset-Dependent Optimal Sharing Strategy: The ablation study demonstrates that the best sharing configuration (sharing keys only, values only, or both) varies across different datasets. For example, sharing only values (CollectiveV) yielded the best AUC/Logloss on MicroVideo, while sharing both (CollectiveKV) was best on KuaiVideo and Ebnerd. This suggests an inherent trade-off between key and value sharing that is not universally optimized by the current CollectiveKV framework, potentially requiring empirical selection per dataset

**Questions:**

NA

---

> ### Author Response · Authors · 2025-11-22
>
> W1. Empirical Determination of User-Specific KV Dimensionality
>
> R1. We thank the reviewer for raising this question. Since the data vary significantly across different recommendation scenarios, the user-specific KV dimension selection differs for each dataset. Moreover, the discrepancies among recommendation models also make it necessary to determine an appropriate user-specific KV dimension for each model individually. Admittedly, this is one of the main limitations of our approach, which we have discussed in the paper. We hope to develop an automatic mechanism for determining the optimal user-specific KV dimension in future work, which motivates us to further extend this research.
>
> At present, in our experiments, we indeed determined this hyperparameter through multiple iterative trials. Although this approach increases the training cost, the overhead is negligible compared to the efficiency gains during inference. Moreover, in practical applications, it is unnecessary to find the exact optimal dimension — empirically testing two or three candidate dimensions within a small range (within 4%) is usually sufficient to achieve performance surpassing the baseline.
>
> W2. Lack of Adaptive Strategy for $d_u$.
>
> R2. We agree with the reviewer that developing an adaptive strategy for determining $d_u$ is an important direction for further improving our method. As noted in R1, although our current approach relies on manual selection, only a few empirical trials within a small range are typically sufficient to identify a good configuration. Designing an automatic mechanism for selecting $d_u$  remains promising future work and is part of our planned research extension.
>
> W3. Inconsistent Robustness of Training Losses
>
> R3. As we explained in R1, the data vary significantly across different recommendation scenarios, so the hyperparameters may need to be tuned for specific dataset, which is a common practice in the recommendation field.
> We want to emphasize that our main goal is to compress the KV cache, while maintaining good performance. Therefore, slight performance fluctuations across different datasets are acceptable.
>
> W4. Dataset-Dependent Optimal Sharing Strategy
>
> R4. We appreciate the reviewer’s observation. Similar to R3, the optimal sharing strategy can also vary due to significant distributional differences across datasets. In principle, one could tune the sharing configuration for each dataset individually. However, in our ablation study, we did not perform such dataset-specific tuning; instead, we used a consistent configuration to ensure fair comparisons and to demonstrate the general effectiveness of our framework.
>
> As we noted in R3, our main contribution is to compress KV cache, so the slight performance fluctuations are acceptable.

---

### Official Review · Reviewer_U1XG · 2025-11-02

**Soundness:** 3
**Presentation:** 4
**Contribution:** 3
**Rating:** 6
**Confidence:** 4

**Summary:**

It is hard to satisfy the latency requirement for Transformer based sequential recommendation models. CollaborativeKV analyzes the collaborative signals in KV caches and divides it into inter-user and intra-user components by SVD. The proposed method achieves better effectiveness and efficiency than state-of-the-art baselines on benchmark video recommendation datasets.

**Strengths:**

--CollaborativeKV improve the efficiency of KV cache and at the same time the recommendation accuracy without tradeoff.

--Practically, it solves the urgent latency problem by low-rank factorization for Transformer based sequential recommendation models.

--The massive reduction in storage overhead has been verified on real-world datasets in Table 1.

**Weaknesses:**

-The proposed method highly relies on the user similarities, which might not be effective for applications with non-overlapping user interests.

-The recommendation accuracy in terms of GAUC is increased by 0.005 or so with the increase of the global KV pool size to 10000 according to Figure5(a). This performance improvement cannot be ignored compared with its improvement in Table 1. So how to derive the conclusion in the last passage of Page 8: …a relatively small global KV pool is SUFFICIENT to capture shared KV information….

**Questions:**

--Though the cache efficiency improvement is obvious, the effectiveness improvement of CollaborativeKV is relatively small.

--According to the choice of benchmark datasets, it is better to change the title to clarify that the proposed method is designed for video recommendation.

---

> ### Author Response · Authors · 2025-11-22
>
> W1. The proposed method highly relies on the user similarities, which might not be effective for applications with non-overlapping user interests.
>
> R1. We thank the reviewer for this insightful comment. Our method is specifically designed for recommendation scenarios, where user collaborative signals naturally exist. We acknowledge that in scenarios with completely non-overlapping user interests, the reliance on user similarity may limit the effectiveness of our approach. However, such cases are extremely rare in practical recommendation applications, where user behaviors usually exhibit sufficient overlap to allow our method to work effectively.
>
> W2. How to derive the conclusion that a relatively small global KV pool is SUFFICIENT to capture shared KV information.
>
> R2. We thank the reviewer for this insightful question.
> Our earlier conclusion was based on the soft increasing trend suggested by the linear fitting results. After incorporating additional experiments with pool sizes below 1000, we observe a clear and substantial performance drop in these settings, indicating that an excessively small pool is indeed insufficient to capture the shared K/V information. We sincerely thank the reviewer again for raising this helpful question.
>
> In addition, to better characterize the overall trend, we replaced the linear fitting with a quadratic polynomial fitting. As shown in the updated Figure 5, this fitting more accurately reflects the non-linear relationship between the pool size and the performance metrics. Further analysis is provided in Section 5.3 of the revised paper.
>
> Q1. Though the cache efficiency improvement is obvious, the effectiveness improvement of CollaborativeKV is relatively small.
>
> R3. We thank the reviewer for the comment. CollectiveKV is primarily designed to reduce cache latency and memory overhead. On this basis, maintaining performance comparable to the baseline is already sufficient. In fact, as our experiments show, our method not only preserves performance but in some cases even surpasses the baseline, demonstrating that efficiency gains do not come at the cost of effectiveness.
>
> Q2. According to the choice of benchmark datasets, it is better to change the title to clarify that the proposed method is designed for video recommendation.
>
> R4. We appreciate the reviewer’s suggestion. Among the three benchmark datasets used in our experiments, two are video recommendation datasets and one is a news recommendation dataset, so our method is not designed exclusively for video recommendation. We selected the two video datasets because they exhibit substantial differences in user behavior patterns and content characteristics, allowing us to conduct comparisons within the video recommendation domain.

---

### Official Review · Reviewer_tHC6 · 2025-11-04

**Soundness:** 3
**Presentation:** 3
**Contribution:** 3
**Rating:** 4
**Confidence:** 4

**Summary:**

This paper introduces CollectiveKV, a novel approach for compressing the KV cache in sequential recommendation systems by leveraging collaborative signals across users. The authors first analyze and identify significant similarities in KV sequences across different users, and through Singular Value Decomposition (SVD), confirm that KV information can be decoupled into largely shareable collaborative information and a small portion of user-specific information. Based on this, they design the CollectiveKV method, which centers on a learnable global KV pool to capture cross-user shared information and a router network for dynamic retrieval of shared KV. During inference, low-dimensional user-specific KV is concatenated with high-dimensional shared KV to drastically compress the cache size.

**Strengths:**

- The motivation is clear. Through data-driven analysis, the existence of collaborative signals in KV caches is demonstrated from the perspectives of similarity and information decomposition, thus leading to the core idea of "decoupling and sharing".
- The paper pioneers the incorporation of cross-user collaborative signals into KV cache compression, distinguishing it from traditional methods focused on individual sequences in LLMs. By decoupling KV information into shared and user-specific components, it establishes a novel paradigm that could be extended to other recommendation scenarios.

**Weaknesses:**

-  The paper notes that the selection of the user-specific KV dimension lacks an automated strategy (as discussed in Section 6) and currently relies on manual setting (e.g., dimensions within 4% of the attention head dimension). This could necessitate iterative experimentation across different datasets or models.
- As shown in Figure 4, increasing the user-specific KV dimension can degrade performance, but the paper does not deeply explore the redundancy threshold.
-  Ablation experiments reveal that the effectiveness of the sharing strategy varies by dataset (e.g., suboptimal results of loss functions on KuaiVideo), suggesting sensitivity to data characteristics like sequence length and task difficulty. This may limit robustness in diverse scenarios, requiring further cross-domain validation.
- The method is effective on various models and datasets, but its performance is affected by data characteristics (such as sequence length), indicating the need for scenario adaptation. Testing on diverse datasets (such as Amazon product data or Spotify sequences) verifies the method's robustness to heterogeneous behavioral patterns.
- Although the paper provides intuitive support through SVD and similarity analysis, the theoretical basis of the sharing mechanism (e.g., the integrity of shared information after decoupling) is not explored in depth. This may affect the trustworthiness of the method in more complex systems.

**Questions:**

- The global K/V pool size is fixed, which determines the upper limit of information capacity. Experiments show that the global K/V pool size has a relatively small impact on AUC. So, does there exist a lower bound for the pool size?
- If a lower bound exists, how would recommendation performance be affected when long-tail user behaviors cannot be adequately covered?

---

> ### Author Response · Authors · 2025-11-22
>
> W1. The user-specific KV dimension selection.
>
> R1. We thank the reviewer for this helpful comment.
> It is true that the user-specific KV dimension currently lacks an automated selection mechanism. As different recommendation datasets vary considerably in behavioral patterns, and different backbone models also exhibit distinct representational characteristics, the appropriate user-specific dimension must be tuned separately for each dataset and model. We acknowledge this as a limitation of our current approach and have discussed it in the paper.
>
> However, we would like to emphasize that automating the dimension selection is not the core contribution of our work. Our contribution primarily lies in firstly demonstrating that shared–specific KV decoupling enables effective KV compression while maintaining or even improving performance, and that this mechanism is robust across models and datasets. The question of automatically determining the optimal dimension is an important but orthogonal direction, which we plan to explore in future work.
> In practice, it is not necessary to find the exact optimal dimension: empirically, evaluating only two or three candidate dimensions within a small search window (e.g., within 4% of the attention head dimension) is usually sufficient to obtain performance that surpasses the baseline.
>
> W2. The redundancy threshold of user-specific KV dimension.
>
> R2. We thank the reviewer for raising this helpful question.
> To further investigate this issue, we conducted additional experiments with several different dimensions and refitted the results using a quadratic polynomial. Please refer to the updated Figure 4 and Section 5.3 for the new results and analysis. The experimental results show that as the user-specific dimension increases to 100, model performance shows a decreasing trend. This suggests that the redundancy threshold may lie near a very small dimension, which is consistent with our analysis in Section 3.2: user-specific information accounts for only a small portion, while most information can be shared across users. Therefore, only a few dimensions are needed to represent user-specific signals.
>
> W3. Robustness in diverse scenarios.
>
> R3. As we explained in R1, the data vary significantly across different recommendation scenarios, so the hyperparameters may need to be tuned for specific dataset, which is a common practice in the recommendation field.
> We agree with the reviewer that further cross-domain validation would strengthen the conclusions. In fact, the three datasets already cover diverse domains: KuaiVideo has much longer average sequence lengths and higher task difficulty than MicroVideo, while Ebnerd represents a different domain (news recommendation), serving as an cross-domain validation.
>
> W4. Testing on diverse datasets.
>
> R4. We thank the reviewer for highlighting the importance of testing on more diverse scenarios.
> We want to emphasize that our main goal is to compress the KV cache, while maintaining good performance. Therefore, slight performance fluctuations across different datasets are acceptable.
>
> Nevertheless, to further verify the robustness of our method under more different behavioral patterns, we additionally conducted experiments on the Amazon product dataset, following the same setup as in the main paper. Using the HSTU model, our method achieves GAUC = 0.8939, AUC = 0.8954, and Logloss = 0.4522, which correspond to improvements of +0.0015, –0.0002, and –0.0004 over the baseline, respectively. This performance is obtained under a compression ratio of 0.039. These results demonstrate that our approach generalizes well to a substantially different domain.
>
> W5. The theoretical basis of the sharing mechanism.
>
> R5. We thank the reviewer for raising the theoretical concern.
> Please refer to Appendix F in the revised paper.
>
> Q1. Lower bound for the pool size
>
> R6. We thank the reviewer for this insightful question.
> To examine whether a lower bound exists, we additionally evaluate multiple settings with pool sizes below 1000. As shown in the new Figure 5 of the revised manuscript.
>
> Based on these observations, we infer that a lower bound does exist. Conceptually, the lower bound corresponds to the minimum pool size required to capture all collaborative information shared across users. The rise–then–fall pattern we observe suggests that this lower bound is likely located near the extremum of the fitted curve.
> This phenomenon is intuitive: once the pool size becomes sufficient to capture all shared collaborative information, further enlarging the representation space introduces additional noise, and an overly large pool may also reduce the routing accuracy of the router.
>
> Q2. Long-tail behaviors
>
> R7. We thank the reviewer for this important question.
> To investigate this issue in depth, we conducted a series of additional experiments. The detailed results and analyses are provided in Appendix G of the revised manuscript.

---

### Author Response · Authors · 2025-11-28

Dear Reviewers,

Thank you again for your valuable feedback. We have submitted our rebuttal and would be very grateful if you could let us know whether any parts of our response require further clarification. We are happy to provide additional details or run supplementary analyses if that would help in your evaluation.

Thank you for your time and consideration.

Best regards.

---

### Meta-Review · Area_Chair_hzCK · 2025-12-08

**Summary:**

The authors study key-value (KV) caches in sequential recommendation. Their findings show that KV sequences for different users share significant amounts of information and could be stored more efficiently by exploiting these similarities. The reviewers find the contribution interesting and novel. Although they criticise a lack of technical novelty (e.g., compared to other ICLR papers), the paper's value lies in the conceptual idea and its solid implementation. The reviewers also point out that there are still many issues with the present solution, e.g., the threshold that differentiates user and collective parts has to be determined manually, however, the paper does not need to solve all problems at once. Instead, it opens an interesting door that may lead to more efficient algorithms in the future.

**Reviewer Concerns:**

Reviewers raised many different concerns, including technical novelty, manual thresholds, number of data sets in empirical evaluation etc rendering the paper borderline after the initial reviews. I believe the authors succeeded in providing additional clarifications and results which effectively pushed the paper across the border.

**Reviewer Scores:**

I believe the authors did well in the rebuttal and clarified at least the most pressing questions raised by the reviewers.

---

### Decision · Program_Chairs · 2026-01-26

Accept (Poster)